

# Effects of swimming before and during pregnancy on placental angiogenesis and perinatal outcome in high-fat diet-fed mice

Xiaofeng Zhu, Weiwei Chen and Haitang Wang

Child Development Research Institute of Jiaxing University, Jiaxing, China

## ABSTRACT

**Background**. We explored the mechanism underlying exercise-mediated placental angiogenesis and perinatal outcome using mouse models.

**Methods**. Three-week-old C57BL/6 female mice were randomly divided into four experimental groups: standard-chow diet (SC), standard chow diet + exercise (SC-Ex), high-fat diet (HFD), and high-fat diet + exercise (HFD-Ex). After 13 weeks of exercise intervention, the male and female mice were caged. Approximately six to seven pregnant female mice from each experimental group were randomly selected for body composition, qRT-PCR, histological, and western blot analysis. The remaining mice were allowed to deliver naturally, and the perinatal outcome indexes were observed.

**Rusults**. The results showed that exercise intervention significantly improved the body composition and glucose tolerance in HFD-fed pregnant mice. The HFD group showed adipocyte infiltration, placental local hypoxia, and villous vascular thrombosis with a significant ($p < 0.05$) increase in the expression of VEGF and ANGPT1 proteins. Exercise intervention significantly elevated the expression of PPAR$\gamma$, alleviated hypoxia and inflammation-related conditions, and inhibited angiogenesis. sFlt-1 mRNA in HFD group was significantly higher than that in SC group ($p < 0.05$). Furthermore, the HFD significantly reduced ($p < 0.05$) the fertility rate in mice.

**Conclusions**. Thus, HFD aggravates placental inflammation and the hypoxic environment and downregulates the expression of PPAR$\gamma$ and PPAR$\alpha$ in the placenta. However, exercise intervention can significantly alleviate these conditions.

# INTRODUCTION

Maternal environment, lifestyle and other factors may alter some functions of the placenta, which may pose potential health risks to the growth and development trajectory of the fetus; in addition, it is an important factor leading to the metabolic diseases during pregnancy (*Furrer & Handschin, 2015*; *Genest et al., 2012*). Pre-eclampsia (PE) is a human pregnancy-specific disease and is the primary cause of maternal mortality, affecting around 2–8% of the pregnancies, worldwide (*Gealekman et al., 2008*). Although the etiology and pathophysiology of PE are unclear, the significance of placenta in PE pathogenesis is well recognized, as removing placenta can treat the clinical symptoms of PE (*Wasinski et al.,*

Corresponding author
Xiaofeng Zhu,
zhuxiaofeng102@126.com

*2015*). Currently, accumulating evidence show that shallow placental implantation in early pregnancy is the primary cause of PE (*De Falco, 2012*; *Gealekman et al., 2008*; *Kim, Song & Park, 2015*; *Nadra et al., 2010*; *Portilho & Machado, 2018*). Furthermore, placental lesions are caused by the imbalance between pro- and anti-angiogenesis. Angiogenesis is strictly controlled by positive and negative regulators, such as VEGF, ANGPT, Prl2c2, Pr17d1, which specifically act on vascular endothelial cells (*Fournier et al., 2002*; *Gealekman et al., 2008*; *Tarrade et al., 2001*).

In addition, metabolic disorders and insulin resistance associated with maternal obesity are important risk factors for PE (*De Falco, 2012*). The maternal obesity is caused by an unbalanced diet and sedentary lifestyle during pregnancy and a maternal BMI >30 is considered as an important risk factor for PE (*He et al., 2014*). Hence, exercise has become a necessary measure to prevent and treat obesity, dyslipidemia, gestational diabetes mellitus and obesity during pregnancy. Exercise can improve mitochondrial function and increase the oxidative metabolism of fatty acids, and thus, it plays an important role in controlling the blood glucose level and reducing the inflammatory responses during pregnancy (*Bishop-Bailey, 2011*). *Genest et al. (2012)* believe that exercise before and during pregnancy can promote the growth and development of placenta by promoting angiogenesis, which may in turn help in the prevention of PE (*Zhang et al., 2017*).

Peroxisome proliferator activated receptors (PPARs) belong to the nuclear receptor superfamily of ligand activated transcription factors, and include PPAR$\alpha$, PPAR$\beta$/$\delta$ and PPAR$\gamma$. They are expressed in endothelial cells and play an important role in the regulation of cell proliferation, vascular proliferation, inflammation and thrombosis (*Wieser et al., 2008*). Additionally, PPAR$\gamma$ has been shown to play an important role in the development of placental vascular system (*Norheim et al., 2014*). Knockout of PPAR$\gamma$ in mice showed early embryonic death due to severe changes in the placental vascular system (*Brosens et al., 2007*) that were related to the imbalance between pro- and anti-angiogenic factors (*Fournier et al., 2002*). Furthermore, multiple studies have demonstrated PPAR$\gamma$ to be a therapeutic target for PE (*Barak et al., 1999*; *Park et al., 2009*; *Schaiff et al., 2007*). In addition,the imbalance between pro- and anti-angiogenic factors is also one of the important pathogenesis of PE. Soluble fms like tyrosine kinase-1 (sFlt-1) and placental growth factor (PIGF) can regulate the function of vascular endothelial cells and affect the integrity and permeability of vascular walls. The study showed that sFlt-1 in serum of PE patients increased, while PIGF decreased. SFlt-1 causes vascular endothelial damage through different mechanisms, leading to the occurrence of PE (*Burchardt, 2018*).

However, exercise and high-fat diet-mediated endogenous expression of PPAR$\gamma$ in maternal placenta, and the underlying mechanism affecting the growth and development of placenta is not clear till date. Our study hypothesized that high-fat diet would aggravate placental inflammation and intrauterine hypoxia in mice, and exercise could significantly improve the expression of placental PPAR and regulate vascular development.

## RESULTS

### Maternal metabolism

The body weight of the maternal mice showed a gradual increase throughout the pregnancy. Before the second week of pregnancy, the body weight of the mice in high fat diet (HFD) and high fat diet + exercise (HFD-Ex) groups was significantly higher ($p < 0.05$) than those in standard chow diet (SC) and standard chow diet + exercise (SC-Ex) groups. However, in the last week of the pregnancy, no significant difference ($p > 0.05$) in the body weight of mice between the groups was observed (Fig. 1A). Furthermore, on the 19th day of pregnancy, body composition analysis demonstrated no significant difference between SC and SC-Ex groups (SC = 4.01 ± 0.6 g, SC-Ex = 3.60 ± 0.64 g). As shown in Fig. 1B, exercise intervention effectively reduced the body fat of mice in the HFD group (HFD = 7.56 ± 1.59 g, HFD-Ex = 6.10 ± 1.41 g, $n = 6$–7, $p < 0.05$). The area under the receiver operating characteristic curve (AUC) of the HFD-Ex group was significantly lower than that of the HFD group, however, it was significantly higher ($p < 0.05$) than that of the SC-Ex group (Fig. 1D). Furthermore, the liver index of the HFD-Ex group was significantly lower ($p < 0.05$) than that of the HFD group, whereas, the difference was insignificant ($p > 0.05$) when compared to that of the SC-Ex group (Fig. 1E).

### Expression of VEGF, ANGPT1, ANGPT2 and sFlt-1 mRNA in placenta

Compared with SC group, the relative expression of VEGF, ANGPT1, ANGPT2 and sFlt-1 mRNA in HFD group was significantly increased ($p < 0.05$). Compared with HFD group, the relative expression of VEGF, ANGPT1, ANGPT2 and sFlt-1 mRNA in HFD-Ex group decreased significantly ($p < 0.05$). Compared with SC-Ex group, there was no significant difference in the relative expression of sFlt-1 mRNA in HFD -Ex group ($P > 0.05$), while the mRNA expression of the other three genes increased significantly ($p < 0.05$), Fig. 2.

### Expression of proteins in maternal mice tissues

The expression of PPAR$\gamma$ was significantly higher ($p < 0.05$) in the HFD-Ex group than that in the HFD group, however, it was significantly lower ($p < 0.05$) than that in the SC-Ex group (Fig. 3C). Furthermore, no significant difference in PPAR$\alpha$ expression was observed between the exercise intervention groups, whereas it was significantly higher ($p < 0.05$) in the SC-Ex group than that in the SC group (Figs. 3E). As shown in 3D, the expression of HIF1$\alpha$ was significantly lower in the placenta of the HFD-Ex group than that in the HFD group, however it was significantly higher ($p < 0.05$) than that in the SC-Ex group. No significant difference ($p > 0.05$) in TNF$\alpha$ expression was observed between the HFD-Ex and SC-Ex groups (Fig. 3F). The placental expression of VEGF and ANGPT1 protein was found to be similar. These proteins were significantly higher ($p < 0.05$) in the HFD group than in the SC and HFD-Ex groups, however, no significant difference was observed in their expression between the exercise groups (Figs. 3G and 3H). Furthermore, the expression of PIGF was significantly higher in the HFD-Ex than in the HFD group, however it was lower than that in the SC-Ex group ($p < 0.05$). Additionally, the expression of PIGF was significantly higher ($p < 0.05$) in the SC-Ex group than that in the SC group (Fig. 3I).

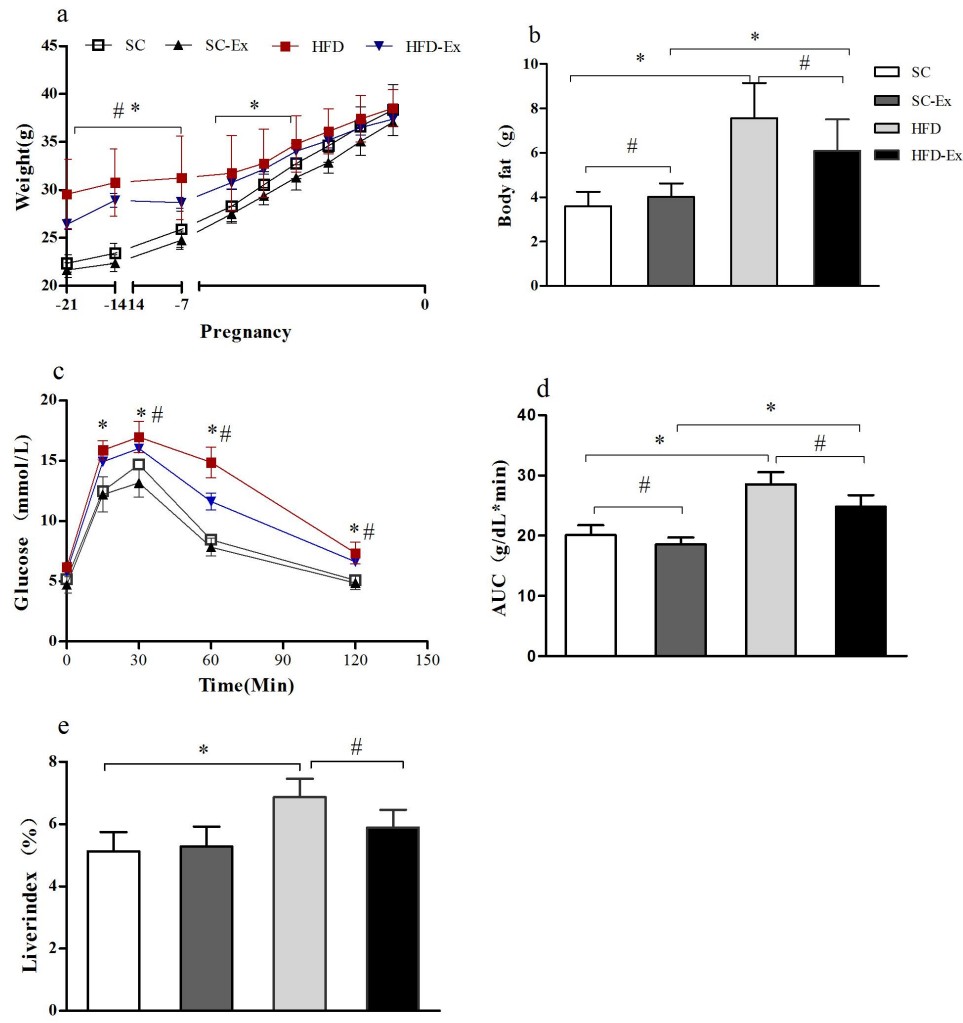

**Figure 1** **Metabolism of maternal mice on the 19th day of pregnancy.** (A) Body weight. (B) Body fat. (C) Glucose tolerance. (D) Area under the receiver operating characteristic curve (AUC). (E) Liver index. $*p < 0.05$ denotes the significant effect of different diets, $\#p < 0.05$ denotes the significant effect of exercise intervention. The body weight was measured on the day of fertilization (F1), monitored once a week in the following two weeks, and monitored daily in the last week.

## Histopathology of placental and white adipose tissues

Hematoxylin and Eosin (H&E) staining of the placental tissues showed diminished spongiotrophoblast, labyrinth and decidual layers in the HFD group (especially the boundary between spongiotrophoblast and labyrinth layer was not clear) compared to the other three experimental groups. Furthermore, vascular dysplasia and thinner decidual layer was observed in the HFD group. Additionally, higher number of erythrocytes in the placental tissue vascular lumen, extravasation of erythrocytes, local clumps and villous vascular thrombosis were observed in the HFD group (Fig. 4A). In addition, fatty cells in HFD group showed larger fat droplets. (Fig. 4B).
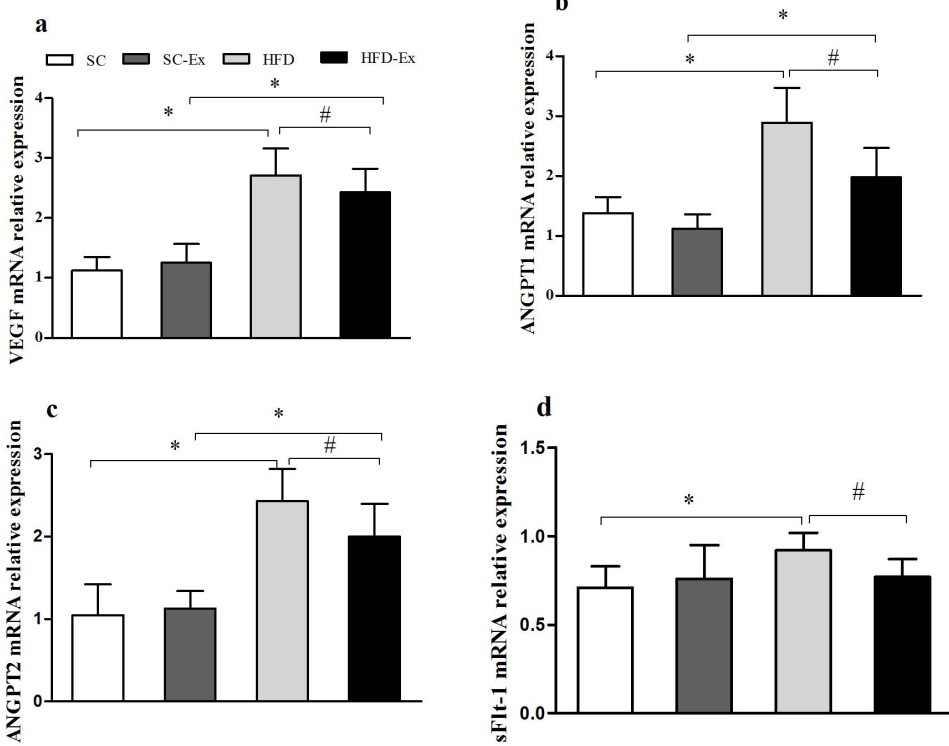

**Figure 2** **Expression of VEGF, ANGPT1, ANGPT2 and sFlt-1 mRNA in placenta.** (A) VEGF mRNA. (B) ANGPT1 mRNA. (C) ANGPT2 mRNA. (D) sFlt-1 mRNA. $*p < 0.05$ denotes the significant effect of different diets, $\#p < 0.05$ denotes the significant effect of exercise intervention.

## Perinatal outcome in maternal mice

The pregnant mice that were not euthanized were allowed to give birth naturally, and the length and body weight of pups, and the fertility rate of mice in each group were observed and recorded. The results indicated no significant difference ($p > 0.05$) in the number of pups and their body weight or length across the experimental groups (Figs. 5A and 5C). Additionally, mice in the HFD group had the lowest fertility rate (45%). Although, exercise intervention improved the fertility rate of these mice to a certain extent (60% in HFD-Ex), it was still lower than that of the mice in SC group (76%) (Fig. 5B). Furthermore, compared to the SC group, pups from the HFD group maternal mice had edema (Fig. 5D).

## DISCUSSION

Placenta is an important organ for the exchange of materials between the fetus and the mother. The placental blood circulation disorder often leads to pathological states, such as intrauterine growth restriction and PE (*Fournier et al., 2002*). Several studies have confirmed that clinical pregnancy specific diseases, including PE, gestational diabetes mellitus and intrauterine growth restriction are associated with the deregulation of PPARs (*De Falco, 2012*; *Forootan et al., 2016*; *Gealekman et al., 2008*; *Park et al., 2009*; *Schaiff et al., 2007*). The role of PPAR$\gamma$ in these diseases is particularly well known (*Park et al., 2009*).

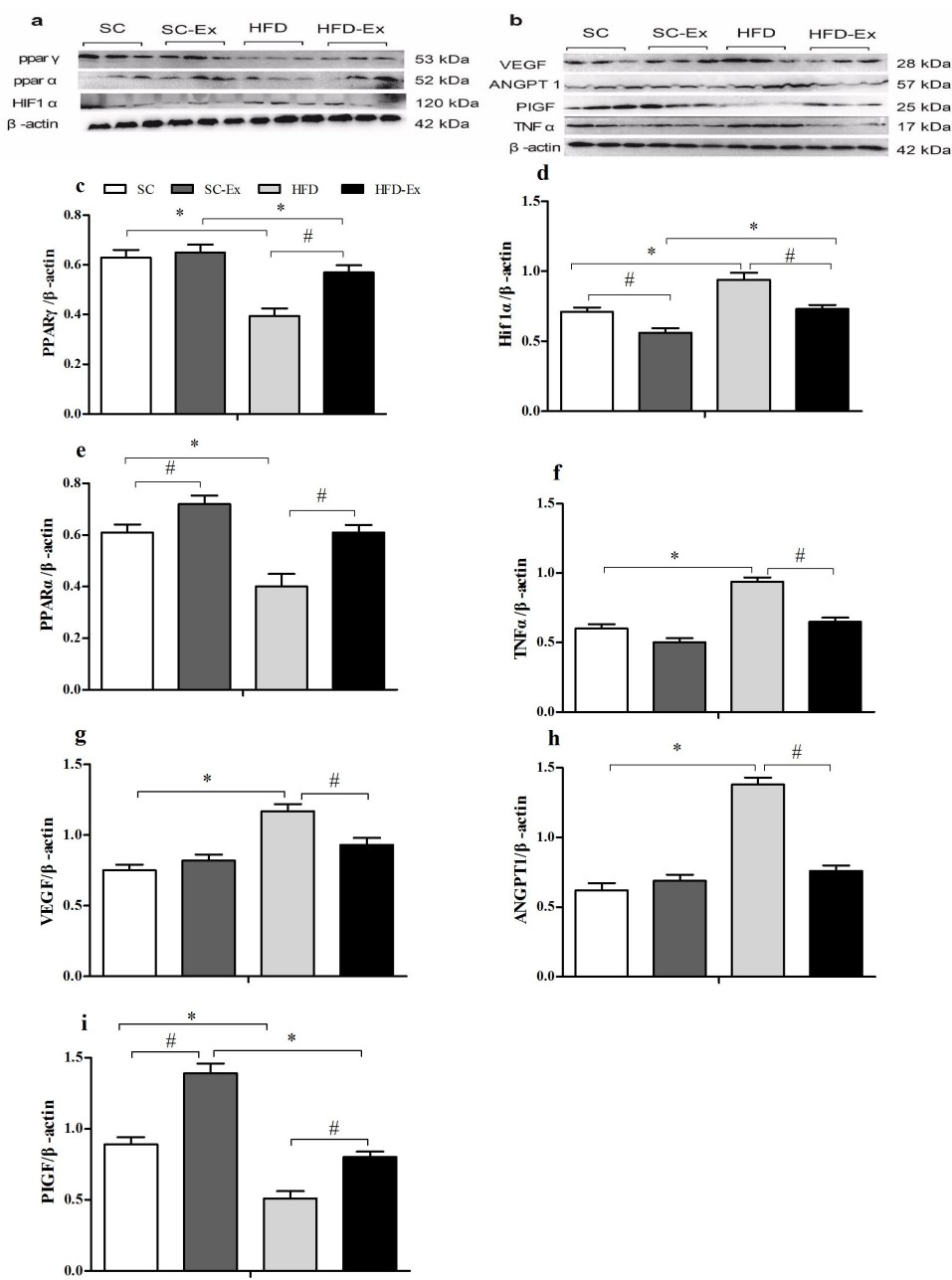

**Figure 3 Expression of proteins in maternal mice fed with different diets.** (A and B) Protein bands. (C) PPARγ/β-actin. (D) Hif1α/β-actin. (E) PPARα/β-actin. (F) TNFα/β-actin. (G) VEGF/β-actin. (H) ANGPT1/β-actin. (I) PIGF/β-actin. *$p < 0.05$ denotes the significant effects of different diets; #$p < 0.05$ denotes the significant effects of exercise intervention.

In early pregnancy, PPARγ is mainly expressed in the invasive trophoblast, whereas in the second trimester of pregnancy, it is mainly expressed in trophoblast cells. In the third trimester of pregnancy, PPARγ is mainly located in extravillous cytotrophoblasts and syncytiotrophoblasts and regulates the production and secretion of placental hormones

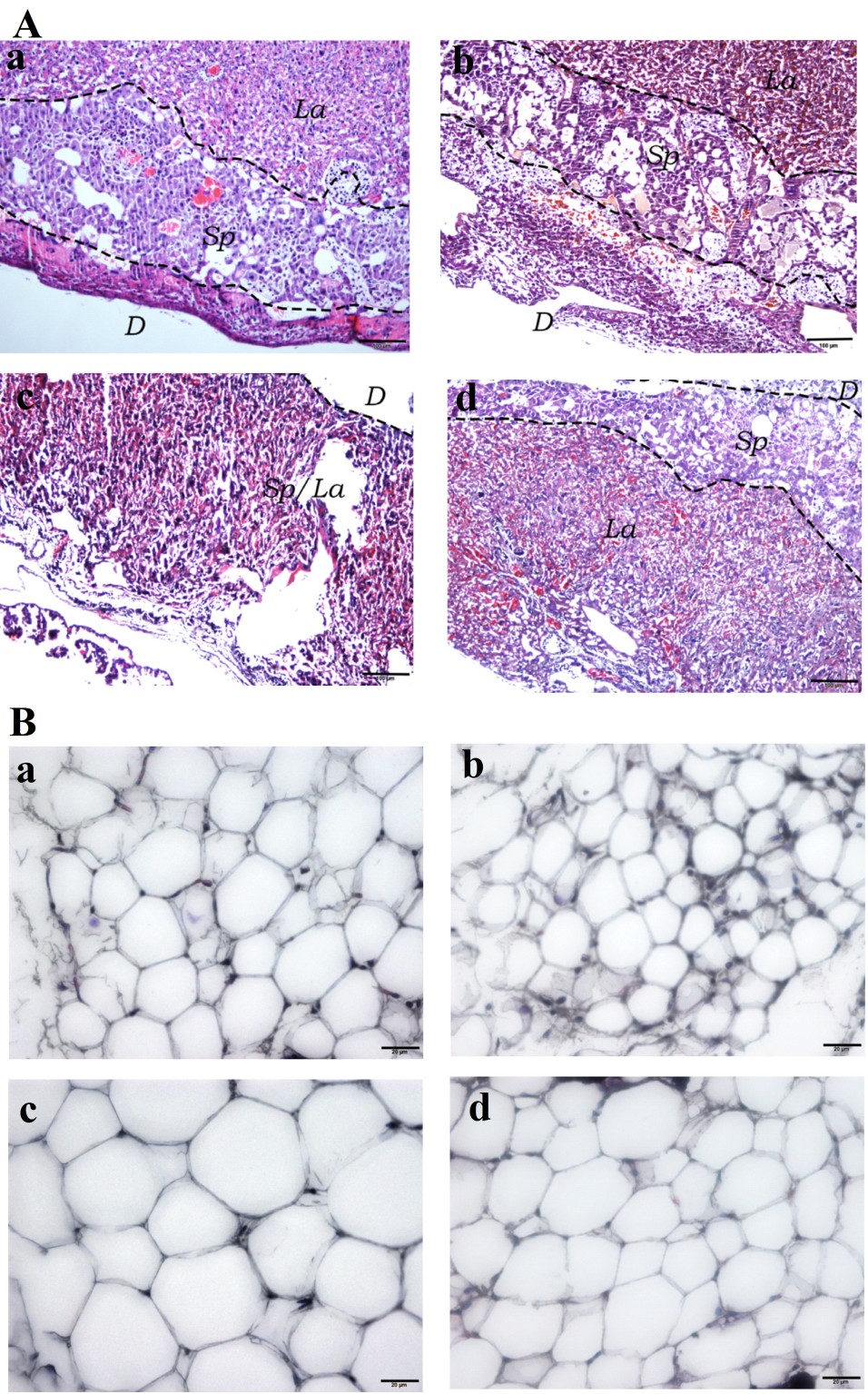

**Figure 4  Histological staining of placental and white adipose tissues.** (A) Hematoxylin and Eosin (H&E) (continued on next page...)

**Figure 4 (…continued)**
staining of placental tissues (40×). (Aa) SC. (Ab) SC-Ex. (Ac) HFD. (Ad) HFD-Ex. (B) H&E staining of adipose tissues (400×). (Ba) SC. (Bb) SC-Ex. (Bc) HFD. (Bd) HFD-Ex. La, labyrinth layer; Sp, spongiotrophoblast layer; D, decidual layer. H&E staining of placental tissue in HFD group showed that the boundary between spongy trophoblast and labyrinth layer was unclear, decidual layer became thinner, the number of red blood cells increased, fatty cells in HFD group showed larger fat droplets.

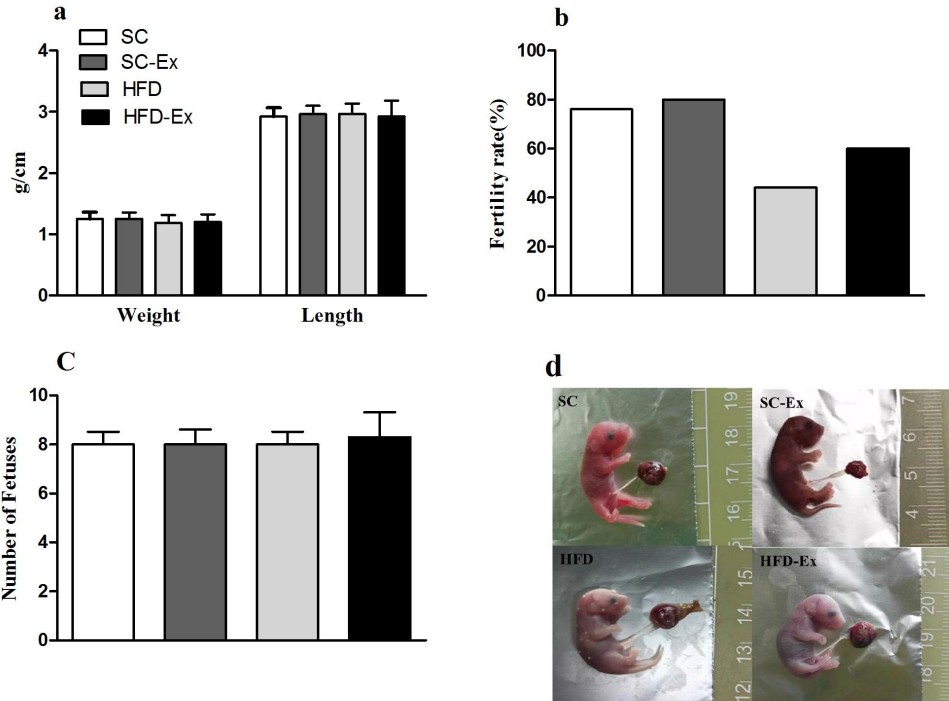

**Figure 5 Perinatal outcome.** (A) Body weight and length of F1 progeny. (B) Fertility rate of female mice (%). (C) Number of pups. (D) Newborn mice, edema was observed in the HFD group. The fertility rate is calculated by dividing the number of normally delivered females by the total number of females in this group. Pups from the HFD group had edema.

(*Lloyd et al., 2005*). PPAR $\gamma$-knockout mice die due to the defect in placental vascular differentiation on the 9.5–11.5th days of the embryonic development (*Brosens et al., 2007*) In fact, PPAR$\gamma$/ RXR $\alpha$ heterodimer has been shown to play a key regulatory role in embryonic development (*Peeters et al. , 2005*). Furthermore, PPAR$\gamma$ plays an important role in embryo implantation. Peeters confirmed that PPAR$\gamma$ ligand reduced the production of endometrial angiogenic factor, VEGF and hypothesized that the associated pathway may affect early embryonic angiogenesis (*Park et al., 2009*). In addition, PPAR$\gamma$ agonists can also promote angiogenesis through the effect on endothelial progenitor cells (*Biscetti et al., 2008*; *Wagner & Wagner, 2020*). Thus, these studies suggest that PPAR is required not only for trophoblast invasion and differentiation, It is also necessary for the development of placental vessels. However, the role of PPAR$\gamma$ in angiogenesis is controversial. Several studies have shown that the activation of PPAR$\gamma$ may inhibit angiogenesis (*Fournier et al., 2002*; *Park et al., 2009*). On the contrary, a few studies have demonstrated the role of

PPAR $\gamma$ in promoting angiogenesis (*Forootan et al., 2016*). PPAR $\gamma$ is known to promote angiogenesis by regulating the expression of VEGF in myocardial tissue and pulmonary capillaries (*Chintalgattu et al., 2007*). However, whether the different roles of PPAR$\gamma$ depends on specific species or cell types is yet to be explored.

PPAR$\gamma$ and PPAR$\alpha$ exhibit anti-inflammatory effects, mainly by inhibiting NFkB, AP-1 and STAT (*Carter et al., 2015*; *Wieser et al., 2008*). Exercise has similar anti-inflammatory effects (*Day et al., 2015*; *Liu et al., 2017*; *Yousefabadi et al., 2021*). In this study, PPAR$\gamma$ protein expression was shown to be significantly increased in mouse placenta under exercise intervention. In addition, TNF$\alpha$ expression was found to be significantly increased in the placenta of pregnant mice from the HFD group.

A significant increase in inflammatory factor, TNF was observed in the placenta of high-fat fed pregnant rats, however, long-term exercise significantly reduced its expression. The upregulation of PPAR$\gamma$ upon exercise may be related to the increase in glucocorticoid secretion (*Wieser et al., 2008*). In addition, PPAR$\alpha$ showed a similar trend; a significant increase in placenta was observed after exercise. Further, a few studies revealed that the abortion rate of PPAR$\alpha$-knockout mice was higher. This could be due to the imbalance between Th1 and Th2 lymphocytes, which may be related to the downregulation of Th1-related transcription factors (*Leite et al., 2017*).The current study also showed low-fertility rate in HFD group, however, exercise partially reversed effects of high-fat-induced infertility. Although we failed to observe significant inter group differences in the body weight and body length of newborn mice, we found that the fertility rate of HFD group was the lowest. In addition to the effects of hormones in the body, there was the possibility of macrosomia in the fetus, which made delivery difficult. However, the newborn mice in HFD group showed edema compared with those in SC group, which may be related to the blood supply of uterus and placenta. With the growth and development of the offspring, this adverse metabolic effect may become more significant.

In recent years, VEGF family has been a research hotspot in many disciplines. Its mechanism is complex and involves many aspects. There have been many studies in cardiovascular field, tumor vascular remodeling and regulating bone marrow hematopoietic function (*Vaz-de Macedo & Clode, 2017*). SFlt-1 and PIGF can regulate the function of vascular endothelial cells. Excessive sFlt-1 in blood circulation is an important anti angiogenic factor that causes hypertension, proteinuria, edema and other symptoms in PE patients, and can cause vascular endothelial damage (*Burchardt, 2018*). PIGF can promote the development of placental vessels, and maintain the normal blood supply and function of the placenta during pregnancy. The synthetic ability of PIGF is weakened, which can reduce the proliferation and infiltration ability of trophoblasts, cause placental ischemia and hypoxia, and lead to disease. In this study, sFlt-1 mRNA in HFD group was significantly higher than that in SC group, But exercise decreased its relative expression in placenta. In addition, sFlt-1 and VEGF showed a analogous effect, which inhibited the biological function of PIGF and caused placental vascular disorders.

Angiogenesis involves the growth of new capillaries in the muscle and other tissues, as a result of endothelial cell proliferation and migration. Although enhanced number of capillaries improves the oxygen transport to individual cells, it does not lead to an

overall increase in muscle blood flow, because of the resistance in the circuit, upstream of capillaries. Although hemodynamic and tissue mechanical tension are behind the increased angiogenesis, ischemia is considered to be the primary stimulant of angiogenesis (*Tarrade et al., 2001*). Angiogenesis is an adaptive physiological response to hypoxia *in vivo* and *in vitro*. Hypoxia inducible factors (HIFs) are the key mediators of angiogenesis and are responsible for activating several angiogenic factors (*Portilho & Machado, 2018*; *Schaiff et al., 2007*). The formation of the vascular network of the placenta is important for the normal supply and exchange of blood, nutrients and oxygen to the fetus. Our histopathological analysis showed an increased number of red blood cells in the placental vascular cavity of HFD mice, suggesting intrauterine hypoxia, red blood cell extravasation and villous vascular thrombosis to a certain extent (Fig. 4A). Furthermore, H&E staining of adipocytes from the HFD group showed fat infiltration (Fig. 4B). Thus, these findings suggest that high-fat diet leads to maternal obesity during pregnancy, chronic inflammation of placenta, and intrauterine hypoxia.

Angiogenesis is regulated by a variety of growth factors, such as angiotensin, VEGF, PIGF, and Angpt1 (*Sassa et al., 2004*; *Seneviratne et al., 2016*). VEGF is a potent vascular permeability factor for endothelial cells and is expressed in villous syncytiotrophoblasts and extravillous cytotrophoblasts. Angpt1 is a vascular-derived growth factor secreted by endothelial cells. It has a similar role to that of VEGF in the development, differentiation and degeneration of blood vessels, and plays a key role in tumor proliferation, invasion, and metastasis (*Sassa et al., 2004*; *Seneviratne et al., 2016*; *Tarrade et al., 2001*). Multiple studies have confirmed the increased serum levels of VEGF and Angpt1 in placental trophoblastic disease and tumor patients, indicating the association of pro-angiogenesis factors with these diseases (*Barak et al., 1999*; *Zhang et al., 2017*). In the current study, VEGF and Angpt1 had a similar expression pattern in the placenta of pregnant mice (Figs. 3G and 3H) with higher levels in the HFD group than in the HFD-Ex and SC groups. However, PIGF expression showed an opposite trend compared to that of VEGF(Fig. 3I). PIGF has an autocrine effect on trophoblast cell function and paracrine effect on blood vessel growth. In addition, it is also a differential indicator for predicting PE. In the current study, exercise before and during pregnancy significantly enhanced the expression of maternal placental PIGF, in both HFD and SC groups.

Furthermore, can these angiogenesis-related phenomena be explained by maternal obesity, low physical activity or metabolic disorder, leading to intrauterine inflammation and hypoxia, followed by the decrease in PPARγ expression and the combined action of PPARγ and HIF1α to activate the angiogenic factors? Is this vascular proliferation a compensatory physiological change or adaptation, and does it increase the risk of macrosomia in offspring, without affecting the body weight and length of the pups? Exercise is known to promote the health of the mother and offspring during the pregnancy (*Bolat et al., 2010*; *Rajia, Chen & Morris, 2013*). However, there are very few reports on improving PPAR γ-mediated placental angiogenesis. The usage of rosiglitazone (a PPARγ agonist) in pregnant mice has been shown to reduce the thickness of the cavernous trophoblast and the surface area of the labyrinth vascular system, and modulate the expression of proteins and accumulation of fatty acids involved in placental development (*Yessoufou et*

*al., 2006*).This is counterproductive. Exercise during pregnancy is a non-invasive treatment for obese pregnant women, which can effectively control weight and improve pregnancy outcomes (*Kubler et al., 2021*). For the way of exercise during pregnancy, researchers believe that swimming is the most appropriate, because water is an effective medium for heat dissipation. The pressure of water can accelerate the blood dynamics in the body, increase the blood volume per unit time, and therefore increase the blood supply of the fetus. In addition, swimming is relatively safe compared with other types of sports, and the sports system does not have to bear the extra load brought by weight gain during pregnancy. However, whether forced swimming is suitable for pregnant mice may have adverse effects on the emotional and spiritual stimulation of animals. In addition, we used 30 min of swimming training during pregnancy. Whether the intensity reached the standard of "Target Heart Rate" or not, it is also necessary to continue to conduct in-depth group controlled trials.

## MATERIALS AND METHODS

### Animals and diet

A total of 130 female and 70 male C57BL/6 mice, aged 3 weeks were purchased from the Shanghai SLAC Animal Center (license No.: syxk 2015-0009). The mice were housed in the Experimental Animal Research Center (SPF level), Shanghai University of Sports, and maintained in an environmentally controlled vivarium with temperature ranging from 65–70°F and a 12-h light/dark cycle. The study was approved by the Animal Experiment Ethics Committee of Shanghai University of Sports (2017040).

After one week of adaptation to SC diet, the female mice were randomly divided into four groups: standard-chow diet (SC, 12% kcal fat, Jiangsu Xietong, China), standard-chow diet + exercise (SC-Ex), high-fat diet (HFD, 45% kcal fat, Research Diets ,USA), high-fat diet + exercise (HFD-Ex). HFD feeding for 16 weeks, there was no special restriction on dietary intake throughout the study. The male mice were fed with standard diet and were not subjected to exercise.

### Swimming exercise intervention

In the current study, we conducted a swimming (phased weight-bearing) exercise intervention on female mice, as described by Wasinsk (*Yancopoulos et al., 2000*). The self-made swimming pool for mice was 50 cm long, 40 cm wide, and 40 cm deep with the water temperature maintained at 30 $\pm$1 °C. Exercise intervention was carried out 13 weeks before pregnancy and 3 weeks during pregnancy. The training frequency was five days a week and the exercise plan was as follows: 10 min in the first week and 20 min in the second week; 10 min was added every week until 60 min when the mice could complete swimming without the load in the 6th week. Subsequently, 3% weight-bearing swimming was started in the 7th week. The initial time was 30 min, and then, 5 min was added every week until 60 min when the mice could complete swimming with weight in the 13th week. The swimming intensity was reduced moderately during cage closing and pregnancy, without load for 30 min each time. The mice in the quiet control group were

immersed in a water tank with the same water temperature and a water depth of three cm, to induce stress caused by water and experimental personnel.

## Mating and pregnancy calculation

After 13 weeks of exercise intervention, male and female mice were caged at a ratio of 1:2. The cage closing duration was 1–5 days, and vaginal suppository was checked daily at 8 am and 5 pm to evaluate the fertilization of female mice. The fertilized female mice were separated from the male mice and fed in a single cage, and the day of fertilization was considered as the first day of pregnancy (F1). The body weight of the female mice during the pregnancy was monitored daily and the animals not showing significant increase in their body weight by the 14th day of pregnancy were excluded from the study (*Salihu et al., 2012*).

## Glucose tolerance test (GTT) and body composition analysis

GTT was performed on the 14th day of pregnancy. After 12 h of fasting, blood was collected from the caudal vein, and glucose was injected at a dose of 1 g/kg (Sinopharm, Beijing, China). The test time points were 0, 15, 30, 60 or 120 min after injection. The AUC was calculated using the Area Below Curve function of SigmaPlot 12.0.

Furthermore, on the 19th day of pregnancy, six-to-seven maternal mice from each group were randomly selected for body composition analysis. After isoflurane inhalation anesthesia, IRIS-CT (Inviscan SAS, Strasbourg, France) was used for whole body scanning to assess various parameters including lean body weight and fat volume. Fat weight was calculated as follows: fat volume $\times 0.95$.

## Tissue extraction

Euthanasia was performed immediately after the analysis of body composition using a 1% Pentobarbital Sodium for intraperitoneal anesthesia. Perirenal fat, liver, fetal mice and placental tissues were immediately removed. The placenta and adipose tissues were collected randomly from each female mouse and fixed in 4% paraformaldehyde solution. Liver tissue was weighed after extraction and liver index was calculated by dividing the weight of the liver by the body weight. Other tissues were frozen in liquid nitrogen following extraction and stored at $-80\ ^{\circ}$C until further use. The blood samples were collected, maintained at room temperature for 30 min, and then serum was separated by centrifugation at 4,000 rpm for 5 min, and stored at $-80\ ^{\circ}$C .

## Determination of angiogenesis gene mRNA in placenta

Total RNA of placenta was extracted by Trizol. Measure the concentration of total RNA and synthesize cDNA after reverse transcription, Determination of VEGF, ANGPT1,ANGPT2, sFlt-1 mRNA expression by qRT-PCR.The internal reference gene was GAPDH, forward primer 5′-GGTTGTCTCCTGCGACTTCA-3′, reverse primer 5′-TAGGGCCTCTCTTGCTCAGT-3′, VEGF forward primer 5′-GCACATAGAGAGAA TGAGCTTCC-3′, reverse primer 5′-CTCCGCTCTGAACA AGGCT-3′, ANGPT1 forward primer 5′-TGCACTAAAGAAGGTGTTTTGCT -3′, reverse primer 5′-CCGGTGTTGTA

TTACTGTCCAA-3′, ANGPT2 forward primer 5′-CCTCGACTACGACGACTCAGT-3′, reverse primer 5′-TCTGCACCACATTCT GTTGGA-3′, sFlt-1 forward primer 5′-GCACCTTGGTTGTGGCTGACT-3′, reverse primer 5′-GGGCCCGGGGGTCTCATTATT-3′. The qRT-PCR amplification reaction condition is set as 50 °C for 2 min ×1 cycle, 95 °C 10 min ×1 cycle, 95 °C 15 s ×40 cycles. GAPDH was used as internal parameter. RQmin, Rqmax and Ct values was automatically analyzed by the software, and $2^{-\Delta\Delta Ct}$ was used for correlation quantification. The Primer BLAST function of PUBMED homepage was used to design relevant primers, and Shanghai Sangon Biotechnology Co., Ltd. (Shanghai, China) was entrusted to synthesize primer sequences.

## Immunoblotting

The frozen placenta was thawed, washed with radioimmunoprecipitation assay buffer (p0013b, Beyotime, Beijing, China), and cut into 2–3 mm pieces, followed by homogenization and ultrasonication. Then, the supernatant was used for the quantification of protein concentration using bicinchoninic acid assay (p0010s, Beyotime, Beijing, China). The protein samples were separated with 8–10% sodium dodecyl sulfate polyacrylamide gel electrophoresis, transferred to polyvinylidene fluoride membrane, washed with TBST buffer thrice for 5 min each time, and then blocked with 5% skimmed milk for 2 h. The samples were then incubated with primary antibodies against PPARγ (CST, #2443, 53 kDa, 1:1000), PPARα (ab3484, 52 kDa, 1:1000; Abcam, Cambridge, UK), ANGPT1 (ab8451, 57 kDa, 1:500; Abcam, Cambridge, UK), VEGF (CY2367, 28 kDa, 1:1000; Abways, Kärnten, Austria), β-actin (AB0033, 42 kDa, 1:5000; Abways, Kärnten, Austria), HIF1α (#36169, 120 kDa, 1:1000; CST, Danvers, MA, USA), PlGF/PlGF (ab196666, 25 kDa, 1:500; Abcam, Cambridge, UK), TNFα (#3707, 17 kDa, 1:1000; CST, Danvers, MA, USA) at 4 °C for 10 h. After washing thrice, the samples were incubated with secondary antibody (1:10000; Abways, Kärnten, Austria) at room temperature for 1 h. The bands were captured using enhanced chemiluminescence detection reagent and Tanon imaging system (Tanon-5200Multi, Shanghai, China).

## Histological analysis of placental and white adipose tissues

White adipose and placental tissues were fixed with 4% paraformaldehyde solution for 48 h and rinsed with running water for 10 min. The tissue blocks were trimmed and the placental tissue was cut along the vertical axis. The tissue sections were then dehydrated with gradient alcohol (50% at room temperature for 12 h, 75% at room temperature for 1 h, 85% at 60 °C for 1 h, twice with 95% at 60 °C for 45 min, and twice with 100% at 60 °C for 15 min), and xylene transparent (100% alcohol and xylene mixture for 30 min, xylene I for 30 min, and xylene II for 10 min). After wax dipping (xylene paraffin solution for 1 h, low wax at 48–50 °C for 1 h, high wax at 56–60 °C for 3 h), paraffin embedding was performed, the tissue sections were marked, cut, and then dried at 37 °C for 12 h. After drying, the tissue sections were dewaxed with xylene and gradient alcohol, washed with running water for 3 min, and then stained with H&E. The images were captured following resin sealing. The borders of the layer were drawn manually using Photoshop CS6 software.

## Statistical analysis

All statistical analyses were performed using SPSS 19.0 software, and the data were expressed as mean $\pm$ standard error. The effects of diet and exercise on female mice were analyzed using two-way analysis of variance and the results with $p < 0.05$ were considered statistically significant. Image J software was used for protein band analysis, and GraphPad prism 5 software was used for analyzing the differences between the glucose tolerant groups.

## CONCLUSIONS

A high-fat diet exacerbated placental inflammation and hypoxic environment and downregulated the expression of PPAR$\gamma$ and PPAR$\alpha$ proteins in the placenta. However, these conditions were significantly improved by exercise. Furthermore, high-fat diet increased the secretion of placental angiogenic factors, which may be a short-term compensatory physiological adaptation. Additionally, our results indicated that high-fat diet and low physical activity reduces the fertility rate in mice, without affecting other perinatal outcomes.

### Funding

This study was supported by the Humanities and Social Science Research Youth Fund Project of Mini-try of Education (20YJCZH253) and Child Development Research Institute of Jiaxing University (20PY3-1). The funders had no role in study design, data collection and analysis, decision to publish, or preparation of the manuscript.

### Grant Disclosures

The following grant information was disclosed by the authors:
Humanities and Social Science Research Youth Fund Project of Mini-try of Education: 20YJCZH253.
Child Development Research Institute of Jiaxing University: 20PY3-1.

### Competing Interests

The authors declare there are no competing interests.

### Author Contributions

- Xiaofeng Zhu conceived and designed the experiments, performed the experiments, prepared figures and/or tables, authored or reviewed drafts of the article, and approved the final draft.
- Weiwei Chen performed the experiments, authored or reviewed drafts of the article, and approved the final draft.
- Haitang Wang analyzed the data, prepared figures and/or tables, and approved the final draft.

### Animal Ethics

The following information was supplied relating to ethical approvals (i.e., approving body and any reference numbers):

The study was approved by the Animal Experiment Ethics Committee of Shanghai University of Sports.

### Ethics

The following information was supplied relating to ethical approvals (i.e., approving body and any reference numbers):

The study was approved by the Animal Experiment Ethics Committee of Shanghai University of Sports (2017040).

### Data Availability

The raw data is available in the Supplementary File.

### Supplemental Information

Supplemental information for this article can be found online at http://dx.doi.org/10.7717/peerj.14562#supplemental-information.

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
