# Peer review of "Effects of swimming before and during pregnancy on placental angiogenesis and perinatal outcome in high-fat diet-fed mice"

_PeerJ, doi:10.7717/peerj.14562_

## Round 0.1 · original submission · Major Revisions

Thank you again for your manuscript submission. The reviewers recommended that you make major amendments to your manuscript. Please revise the manuscript according to the referees' comments.

Reviewer 1 ·

Basic reporting

In the current manuscript, Zhu et al. claim that “Effects of swimming before and during pregnancy on
placental angiogenesis and perinatal outcome in high-fat diet-fed mice”. The authors conclude that long-term (13 weeks) HFD aggravates placental inflammation (Figure 2; f, h) and the hypoxic environment (Figure 2; d, g) and downregulates the expression of PPARs (Figure 2c, d) in the placenta. Exercise interventions can also significantly alleviate these conditions (Figure 2) and low fertility rates caused by HFD (Figure 4).

Experimental design

The authors show the effect on angiogenesis by exercise as shown in Figure 2 (several protein levels) and Figure 3A (histology). I suggest that the authors form conclusion from additional evidence rather than forming conclusions solely from Western blot AND poor H&E image. For example, the authors can perform qRT-PCR or RNA-seq analysis for more genes that may support angiogenesis and inflammation. Also, the authors should provide high-resolution H&E histology data to clarify whether exercise affects placental angiogenesis and inflammation.

Validity of the findings

The conclusion that the exercise affects placental angiogenesis under HFD is not convincing, because of insufficient data. The authors should provide clearer or additional data to clarify whether swimming affects placental angiogenesis and inflammation. Otherwise, the authors need to modify their conclusion about the placental angiogenesis in the title and abstract of the present manuscript.

Additional comments

1. Overall, figure legend is very short. Please explain them in detail.
2. The 13 weeks can be relatively short or long term. Please delet ‘long term”.
3. In Figure 1c, the significance with p-value is missing.
4. In Figure 4b, the error bar is missing.

Reviewer 2 ·

Basic reporting

Please, structure the manuscript according to the journal´s standard.

Introduction: The first two paragraphs give the idea that preeclampsia has a central role in the study. I suggest that authors modify the introduction section so readers have a better foundation to understand the purpose of the study.

Experimental design

Please, make the study hypothesis clear in the introduction section

Methods:
- I suggest inserting a figure with the protocol timeline;
- (lines 226-229): It is unclear how long the female rats were on the high-fat diet;
- (lines 235-236): “The period of intervention was 16 weeks: 13 weeks before delivery, 3 weeks during pregnancy, and no exercise during lactation”. Please rewrite the sentence because it is confusing.
- (line 263): I suggest separating the method of euthanasia and tissue extraction from immunoblotting into different topics

Validity of the findings

no comment

Additional comments

The main findings of the present study are that swimming before and during pregnancy prevents inflammation, a hypoxic environment, and the decreased expression of PPARγ and PPARα proteins in the placenta from high-fat-fed rats. Although the study is original research that falls within the scope of the journal, some points need to be improved, as follows:

General: Please, structure the manuscript according to the journal´s standard.

Introduction: The first two paragraphs give the idea that preeclampsia has a central role in the study. I suggest that authors modify the introduction section so readers have a better foundation to understand the purpose of the study. Please, make the study hypothesis clear.

Methods:
- I suggest inserting a figure with the protocol timeline;
- (lines 226-229): It is unclear how long the female rats were on the high-fat diet;
- (lines 235-236): “The period of intervention was 16 weeks: 13 weeks before delivery, 3 weeks during pregnancy, and no exercise during lactation”. Please rewrite the sentence because it is confusing.
- (line 263): I suggest separating the method of euthanasia and tissue extraction from immunoblotting into different topics

Results:
- Figure 1: Graph (a) is not self-explanatory. I could not understand the weight measurement period. I suggest including weight and food intake data from the start of the HF diet.
- Please provide results on lean body weight, according to the Material and Methods sections (IRIS-CT – body scanning).

Discussion:
- I suggest carefully revising the discussion section because the reasoning is difficult to follow.
- I suggest discussing maternal weight data before and during pregnancy and data on maternal metabolism.
- I suggest that authors insert the discussion on the benefits of the specific exercise protocol used in this work.
- The number of fetuses and their weight and length were not different between groups despite long-term high-fat diet exacerbated placental inflammation and hypoxic environment. I suggest that authors discuss these results. Even if there are no significant fetal changes, could there be implications for adult life? I think this discussion is relevant.

---

## Round 0.2 · Minor Revisions

In general, the authors presented new results, satisfactory answers to the demands of the reviewers and new discussions, which improved the quality of the article. I indicate in red and in text boxes some issues that need to be reviewed (attached), and I consider that the article needs a careful proofreading for publication.

Reviewer 1 ·

Basic reporting

In the revised manuscript, Zhu et al. have made a good attempt at answering and added new experimental data to clarify my suggestions and concerns in the initial review. They included new qRT-PCR data and better quality H&E images in the revised manuscript. This revised manuscript provides interesting points about the effect of exercise on placental angiogenesis in HFD. In my opinion, with careful proofreading and typo/spacing corrections, this manuscript can be published in PeerJ.

Experimental design

None.

Validity of the findings

None.

Additional comments

None.

Reviewer 2 ·

Basic reporting

no comment

Experimental design

no comment

Validity of the findings

no comment

Additional comments

The authors have improved the manuscript and I am overall satisfied with their responses.

---

## Round 0.3 · accepted · Accept

The authors have responded to all the requests of the reviewers and the editor, which has improved the quality of the article. I consider the article approved.